# Characterizing Interstellar Medium, Planetary Surface and Deep Environments by Spectroscopic Techniques Using Unique Simulation Chambers at Centro de Astrobiologia (CAB)

**DOI:** 10.3390/life9030072

**Published:** 2019-09-10

**Authors:** Eva Mateo-Marti, Olga Prieto-Ballesteros, Guillermo Muñoz Caro, Cristobal González-Díaz, Victoria Muñoz-Iglesias, Santos Gálvez-Martínez

**Affiliations:** Centro de Astrobiología (CSIC-INTA), Ctra. Ajalvir, Km. 4, 28850-Torrejón de Ardoz, Spain

**Keywords:** simulation chambers, habitability, spectroscopy, interstellar medium, icy moons, planetary surfaces

## Abstract

At present, the study of diverse habitable environments of astrobiological interest has become a major challenge. Due to the obvious technical and economical limitations on in situ exploration, laboratory simulations are one of the most feasible research options to make advances both in several astrobiologically interesting environments and in developing a consistent description of the origin of life. With this objective in mind, we applied vacuum and high pressure technology to the design of versatile simulation chambers devoted to the simulation of the interstellar medium, planetary atmospheres conditions and high-pressure environments. These simulation facilities are especially appropriate for studying the physical, chemical and biological changes induced in a particular sample by in situ irradiation or physical parameters in a controlled environment. Furthermore, the implementation of several spectroscopies, such as infrared, Raman, ultraviolet, etc., to study solids, and mass spectrometry to monitor the gas phase, in our simulation chambers, provide specific tools for the in situ physico-chemical characterization of analogues of astrobiological interest. Simulation chamber facilities are a promising and potential tool for planetary exploration of habitable environments. A review of many wide-ranging applications in astrobiology are detailed herein to provide an understanding of the potential and flexibility of these unique experimental systems.

## 1. Introduction

To characterize the potentially habitable environments of our solar system and beyond is a priority goal in the exploration programs of the main national space agencies around the world. The specific investigations and harsh conditions of such environments are challenging for astrobiology missions, which have to deal with technical and economic issues that constricts the payload working on extreme conditions, and difficulties to establish detection limits low enough for the unequivocal identification of the habitable requirements and detection of biosignatures. Simulation experiments may help with these factors, reproducing a wide range of extreme conditions, from those of the interstellar medium to the irradiated planetary surfaces or confined water reservoirs.

Silicate and carbon dust grains are produced in the atmospheres of asymptotic giant branch (AGB) and expelled to the interstellar medium. The interstellar medium, namely the space between stars, contains dust particles and about 200 different gas-phase species have already been identified. Dense interstellar cloud interiors, with densities above 10^4^ particles cm^−3^, are preserved from the external ultraviolet (UV) field emitted by massive stars, and temperatures below 20 K allow accretion of volatiles on dust surfaces forming ice mantles. Based on IR observations, we know that interstellar ice is made of water and other species including CO, CO_2_, CH_3_, OH, NH_3_, etc. Dense clouds are exposed to the secondary UV flux generated by cosmic-ray excitation of molecular hydrogen. Based on laboratory simulations, UV light leads to the formation of new molecules in the ice mantles; many of these species are of astrobiological interest. Cold circumstellar regions also contribute to icy grain processing due to radiation and warm-up. Coagulation of grains into planetesimals and cometesimals protects these ice photoproducts from subsequent irradiation by cosmic rays, UV photons, and X-rays. These molecules are preserved this way in their parent body. 

The gas and dust in the interstellar medium at different densities condense forming diffuse and dense clouds. In dense clouds, dust grains favour the formation of new stars by absorbing the excess energy generated during the gravitational collapse and emit this energy at infrared wavelengths where the cloud is optically thin, therefore cooling the gas. Conservation of angular momentum induces the appearance of a disk around the protostar, where planets, comets and asteroids are eventually formed (Figure 1). Whether planets accumulate enough energy by accretion, radiogenic heat or tidal forces, some geological processes may produce internal differentiation, resurfacing, and even conspicuous aqueous environments. How do all these processes converge to emerge potential habitats is a difficult question to answer [1].

The limited knowledge that we possess concerning planetary environments has been gathered primarily from space missions. Space missions to comets, asteroids, moons and planetary surfaces dig into the formation of our solar system, providing information to characterize the most primitive matter present in these objects. The delivery of water and exogenous organic materials via comets, (micro-) meteorites and interplanetary dust particles to the primitive Earth likely contributed to the development of prebiotic chemistry. Rosetta, for instance, revealed breaking data concerning the water and the chemical building blocks in the solar system, or the shaping of comets.

There has been an exponential increase in interest among international space agencies (National Aeronautics and Space Administration (NASA), European Space Agency (ESA), Russian Space Agency Roscosmos, and Indian Space Agency) to explore planetary surfaces. Among them Mars and icy moons are crucial targets. A common research goal of these missions is to study habitability conditions, including the detection of atmospheric traces of gases such as methane and unequivocal detection of organics on the surface. The NASA-Mars Science Laboratory (MSL) mission has been operating on Mars for several years now [2]. The European mission ExoMars, with the Trace Gas Orbiter (TGO) and Schiaparelli lander, is used as a precursor mission for the 2020 ExoMars (Rosalind Franklin) rover and surface platform [3]. Both surface missions target, among other topics, habitability conditions studies. The interest on ocean worlds is evident since in the next decade two missions will visit the Jupiter system with the aim to characterize the habitability of deep environments for the first time: JUpiter Icy Moons Explorer (JUICE), led by the ESA [4], will focus on Ganymede while Europa Clipper, led by NASA [5], will fly over Europa.

Although space missions provide fundamental, unique and basic knowledge for space exploration, they are always costly and extremely time-consuming. Due to the obvious technical and economical limitations of space exploration, laboratory simulations are among the most feasible research options for making advances both in space and planetary science, and in developing a consistent description of the origin of life. Therefore, laboratory simulations of several environments (interstellar medium, planetary surfaces and interiors) are a necessary and complementary option to understand data delivered by expensive space missions. Simulation chambers are economical, versatile and allow for a high number of experiments helping to interpret space missions.

Simulation chambers have experienced extraordinary improvement from the initial operational models and instrumental designs in the 1960s to the present [6]. The majority of planetary simulation chambers simulate physical parameters at the surface or atmosphere of a planet [7,8], exceptionally liquid environments [9] or implemented by spectroscopy techniques for in situ sample characterization [10].

In this article, we will present experimental simulations devoted to mimic the processes of astrobiological interest that take place in inter- and circumstellar icy dust grains, the interior of icy moons of the solar system and the surface of planets like Mars. These experiments have been performed in the Laboratory for Simulation of the evolution in Interstellar and Planetary environments (LSAIP: *Laboratorio para Simulación de la Evolución de Ambientes Interestelares y Planetarios*), where there are several facilities with the capability to reproduce specifically interesting environments of the universe. They are open to the science community for collaborative investigations in the topics related to astrobiology. So far collaborations have produced interesting results and links between several international institutes and CAB. 

Three simulation chambers are noteworthy, InterStellar Astrochemistry Chamber (ISAC), Planetary Atmospheres and Surfaces Chamber (PASC) and High-Pressure Planetary Environment Chamber (HPPECs), since they cover together all the planetary environments of the solar system. Their principal strength over other simulation chambers is the incorporation of particular in situ analytical techniques, which allow measurements comparable to those of space missions.

## 2. Materials and Methods 

### 2.1. ISAC

The InterStellar Astrochemistry Chamber (ISAC, Figure 2) is an ultra-high-vacuum (UHV) setup with a base pressure of 2.5–4.0 × 10^−11^ mbar. ISAC was specifically designed to grow a thin ice film of composition analog to interstellar ice mantles. This is achieved by depositing a gas mixture onto a substrate window placed at the tip of a cold finger at 8 K. Ice samples can be UV-irradiated using a microwave stimulated hydrogen flow discharge lamp (MDHL) that provides a flux of 2.5 × 10^14^ photons cm^−2^ s^−1^ at the sample position with an average photon energy of 8.6 eV. The emission of the MDHL simulates the secondary UV field in dense clouds generated by cosmic ray excitation of H_2_ molecules. The MDHL emission spectrum is monitored during the experiments with the use of a McPherson 0.2 m focal length UV monochromator (model 234/302) with a photomultiplier tube (PMT) detector that operates in the 100–500 nm (11.27–2.47 eV) range [11,12,13,14]. The UV absorption spectrum of ice samples of different composition is obtained as the difference of the cold substrate spectra before and after ice deposition [11,12,13]. In addition, a Ni-mesh measures the UV photocurrent and serves to estimate the total value of the UV flux [15].

Temperature-programmed desorption experiments are performed at a constant heating ramp of typically 1 K min^−1^. The ice is monitored by in situ transmittance Fourier transform infrared (FTIR) spectroscopy (Bruker VERTEX 70, Billerica, Massachusetts, USA), while the volatile species are recorded with a quadrupole mass spectrometer (QMS) equipped with a Channeltron detector (Pfeiffer Vacuum, Prisma QMS 200, Asslar, Germany ). For a detailed description of ISAC, we refer to Muñoz Caro et al. (2010) [16].

### 2.2. PASC

PASC (Planetary Atmospheres and Surfaces Chamber) is an ultra-high-vacuum (UHV) simulation chamber 500 mm long by 400 mm diameter that is capable of reproducing atmospheric compositions and surface temperatures that are representative of most planetary objects (Figure 3). This equipment was developed to make feasible the in situ UV irradiation and infrared spectroscopy characterization of samples under study (see the details provided in [17]). The total pressure range of the chamber is from 7 mbar to 5 × 10^−9^ mbar, and the partial pressures of the gases in the chamber can be set with this precision. To simulate a particular atmosphere, the gases are mixed in a many-fold to the required proportions, and the gas composition is constantly monitored by a residual gas analyser mass spectrometer. The temperature of the sample is regulated by a commercial helium cooling system connected to the sample holder, and it can be adjusted from –260 °C to 50 °C. Sample size ranges are from 5 mm to 35 mm in diameter; however, other sizes of sample container are available for specific developments. Various irradiation sources are available to mimic electron (5 KeV), ion (5 KeV) and UV–VIS irradiation (200–500 nm). Furthermore, infrared and RAMAN spectroscopies, in situ analytical techniques expand the powerful characteristics of this planetary environment simulation chamber.

### 2.3. HPPECs

Three high-pressure temperature-controlled chambers (Figure 4), to study samples of high volume (67, 10 and 2 mL), have been designed to work up to 500 (MPPEC (Medium-Pressure Planetary Environment Chamber)), 3000 (HPPEC (High-Pressure Planetary Environment Chamber)) and 10,000 bar (UHPPEC (Ultra-High-Pressure Planetary Environment Chamber)). The three chambers, made of 304 stainless steel, have connections for introducing a pressure sensor and a thermocouple in direct contact with the sample to monitor thermodynamic data. The chambers are connected, through a circuit of high-pressure stainless-steel tubing and valves, simultaneously to both, a hydraulic compressor (Ruska 7615, Ovredal, Madrid, Spain) which allows to pressurize the sample with a fluid (water or oil) up to 3000 bar, and to a gas tank (CO_2_ or CH_4_ bottles in the case of our experiments). The compressor is coupled to a piston that multiplies the pressure by a factor of 10 in the case of the UHPPEC. 

The system is thermostated by the cooling baths Unistat 815w (Huber, Berching, Germany) and Circulator rf/heat adv. Progr. 15L (VWR, Radnor, PA, USA) that work in a temperature range from 203 to 323 K and 233 to 323 K, respectively.

All three chambers have a port closed by a synthetic sapphire window in order to carry out in situ visual and Raman spectroscopy measurements. The Raman used to perform the measurements is the Horiba JobinYvon HRi550 monochromator, with a charge-coupled device (CCD) with 1024 × 256 pixels cooled to 203 K for thermal noise reduction, prepared to work with four diffraction gratings of 600, 1200, 1800 and 2400 grooves/mm which provide a wide range of intensity signal/resolution ratio. Excitation of the sample is done by an intensity-modulated (0–200 mW) Nd:YAG solid state laser with a wavelength of 532 nm non polarized. Fibre optics can connect the spectrometer to different optical devices, which are two cryogenic probes with focal lens of 10 and 22.6 mm (Microbeam S. A., Barcelona, Spain) and to a B and W Tek microscope with a 20× objective (Microbeam S. A., Barcelona, Spain).

## 3. Review of Experimental Results

### 3.1. Interstellar Medium

Since the inauguration of the LSAIP in 2008, ISAC was calibrated and optimized to perform experiments aiming to study the processes that take place in bare carbonaceous dust, and more frequently, icy grain mantles in interstellar clouds. Our laboratory simulations of astrophysical ice submitted to UV/X-ray/swift heavy ion irradiation and warm-up, in collaboration with several national and international institutions, allow the interpretation of ice observations in the infrared and observations of the surrounding gas-phase molecules with radio telescopes. In addition, the molecules synthesized in energetically processed ice analogues are good candidates for their detection toward cold space environments. 

Ice mantles contain mainly water and other volatile species such as CO, CO_2_, CH_3_OH, NH_3_, etc. [18]. Ice irradiation followed by warm-up leads to a plethora of molecules of astrobiological interest that include amino acids, sugars, N-heterocycles, carboxylic acids, etc. These molecules are preserved in comets and other primitive bodies of our solar system, since they were made by agglomeration of tiny icy dust particles. During the epoch of heavy bombardment, cometary and asteroidal impacts delivered their organic loads to the primitive Earth, therefore seeding our planet with prebiotic molecules that might have contributed to trigger the molecular evolution that ultimately led to the first life forms. A summary of our activity performed with the ISAC setup is summarized below.

Our collaboration with the group of E. Dartois at the Institut d’Astrophysique Spatiale (IAS) and Institut des Sciences Moléculaires d’Orsay (ISMO) (France) employs the French facilities and ISAC to simulate in more detail the processes that occur in interplanetary dust particles and bare interstellar carbon grains observed in the diffuse medium [19,20,21,22,23,24,25,26,27,28,29,30,31]. These carbon grains are composed of hydrogenated amorphous carbon and comprise an important fraction of the carbon budget in galaxies including our Milky Way. As mentioned above, the synthesis of relatively complex molecules in icy dust grains, many of them of astrobiological significance, could explain the observed species in the gas phase toward cold regions of space. However, a non-thermal desorption is necessary to release ice molecules to the surrounding gas. Along with direct cosmic ray impact on ice mantles [32], energetic UV and X-ray photons contribute to eject molecules from the irradiated ice. The first publication of the results obtained with ISAC describes this setup and a study of carbon monoxide (CO) ice UV photodesorption, i.e., the release of CO ice molecules to the gas phase due to absorption of UV photons in the ice. We found that only the photons absorbed in the top five ice monolayers participate in the photodesorption [16], where one monolayer has an average thickness of one molecule or a column density of 1 × 10^15^ molecules cm^−2^. We discovered a new type of desorption that was named “photochemical desorption”. This phenomenon leads to a constant desorption rate of photoproducts that are formed on the ice surface and desorb due to the excess energy of the reaction [33]. We also found the first evidence of excitons in ice exposed to UV irradiation [34]. The observation that only the photons absorbed in the top CO ice monolayers lead to a photodesorption event and the high value of the photodesorption rate suggested that the photodesorption quantum yield, expressed in molecules per absorbed photon, should be higher than unity. For their estimation, we were the first laboratory that developed vacuum-UV spectroscopy of ices using the hydrogen lamp with no need to use a synchrotron light source. We measured the absorption cross-sections of several ice species for the first time, which allowed to estimate the efficiency of photodesorption per absorbed photon in the ice [11,12,13]. We extended this study of CO ice UV-photodesorption to other molecular ice components, and started to explore the effect of soft X-ray irradiation in the photodesorption of ice analogues [14,15,33,34,35,36,37,38,39,40,41,42].

Our work with ISAC and other ice setups in Taiwan and Palermo (Italy) has, thus, served to obtain a deeper understanding of ice photochemistry and photodesorption. This study also explored the sulphur chemistry in the ice [14,43,44,45], which led to new observing proposals of sulphur-bearing molecules and to the first detection of HS_2_ in space [46,47]. Our ongoing collaboration with the Palermo Observatory in Italy and the National Central University in Taiwan is devoted to study the effect of X-rays emitted by young stars in the ice. These experiments were conducted in the ISAC setup using a Palermo home-made X-ray source, and later at the National Synchrotron of Taiwan (NSRRC) to increase the flux of soft X-rays [40,44,48,49,50,51,52,53,54,55]. Meanwhile, we employed swift heavy ions provided by the French accelerator GANIL at Caen to reproduce cosmic ray impact in the ice [56]. With few exceptions, we discovered that both ions and X-rays generate products that are common to UV processing of realistic ice analogues [55,56], although their relative abundances can differ significantly depending on the type of energy source. 

Other results obtained with ISAC participation also served to improve our understanding of astrophysical ices. The synthesis of HNCO isomers by ice processing was compared to observations of these molecules in astrophysical environments [57]. We performed a novel spectroscopic study of CO_2_ ice which led to conclude that pure solid CO_2_ in the amorphous state has never been observed in ice mantles [58]. Far infrared spectroscopy of ice analogues was carried out to extend ice observations in this range with the advent of new missions like Space Infrared Telescope for Cosmology and Astrophysics (SPICA) [59,60]. A 3.4 micron feature profile characteristic of organic refractory matter of astrobiological relevance, made by irradiation and warm-up of ice analogue mixtures, was proposed as a tracer of organic material in processed ice mantles in space [61]. Indeed, so far the main form of solid carbon detected in interstellar dust grains, IDPs, and also the bulk carbonaceous chondrites, is dominated by various forms of a more or less hydrogenated amorphous carbon, with only small amounts of biomolecules. Nevertheless, the more pristine cometary nuclei could still host organic matter of astrobiological interest. In this regard, we ran a series of experiments to mimic the thermal desorption of cometary ice, predicting that volatiles would be readily detected far from the Sun by the Rosetta mission [62,63]. Despite the non-nominal landing on the 67P comet nucleus, our long-standing participation in the COSAC-Rosetta instrument on board the Philae landing module, was fulfilled with the detection of complex organic molecules on the surface of a comet [64,65,66], which are compatible with a synthesis in irradiated pre-cometary icy grains, prior to their agglomeration toward the formation of cometesimals in the solar nebula.

For an extended summary of the ice research presented in this section, we refer to [55,67] and the book entitled “Laboratory Astrophysics” [68] with contribution from several experts in this field. 

### 3.2. Planetary Surfaces

PASC simulation facility was developed to be a useful tool in several scientific and technological areas of knowledge. PASC has produced significant scientific outcomes related to astrobiology research due to its versatility design and capability to simulate atmospheric composition and surface temperatures for most of the planetary objects. The near-UV irradiation of basalt dust on Mars has been simulated experimentally to determine the transmittance as a function of the mass and thickness of the dust. This data can serve to quantify the absorption of dust deposited on sensors aiming to measure the UV intensity on Mars surface [69]. 

The presence of perchlorates salts on the Mars surface in not only relevant because of their ability to lower the freezing point of water, but also because they can absorb water vapour and form a liquid solution (deliquesce). We show experimentally at PASC that small amounts of sodium perchlorate (1 mg), at Mars atmospheric conditions, spontaneously absorb moisture and melt into a liquid solution growing into 1 mm liquid spheroids at temperatures as low as 225K. Our results indicate that salty environments make liquid water to be locally and sporadically stable on present day Mars [70]. These results suggest a water cycle on Mars in which atmospheric vapour deliquesces salt particles into liquid solutions, freezes and sublimates. The deliquescent properties of perchlorate salts may, thus, play a significant role in controlling soil and atmospheric water content. The probable existence of liquid saline water on the surface of Mars has important repercussions for the interpretation of Martian data, understanding of local water cycles, the selection of landing sites for future missions and planetary protection. 

Minerals can be very promising surfaces for studying biomolecule surface processes, which are of principal relevance in the origin of life and a source of chemical complexity. Minerals such as silicates, oxides and sulphides were probably present on early Earth in several environments. Therefore in this context we have study interaction of amino acids [71,72,73], small peptides [74,75] and nucleic bases [76] on several mineral surfaces. We have performed the first spectroscopic characterization of triglycine adsorption on a UV irradiated pyrite surface. X-Ray Photoemission Spectroscopy (XPS) analysis was employed to efficiently explore the molecular adsorption, to understand surface chemistry and, finally, to describe the critical influence of the different environmental conditions in the small peptides-pyrite system. We demonstrated how pyrite minerals can concentrate and act as adsorption substrates for small peptides during wetting conditions and even under inert UHV conditions. The successful adsorption of several molecules from amino acids to small peptides under a wide range of experimental conditions indicates the potential of pyrite as a catalyst under different environmental conditions [75]. We have also exposed the adenine surface system to UV radiation (200–400 nm) under a high-vacuum environment (10^-7^ mbar) to study the photostability and photochemistry of adenine on different surfaces. After 10 or 24 hours of exposure under interplanetary space conditions, UV radiation induces desorption and partial dissociation of the molecule, which is dependent on the nature of the surface. Those results reinforce the idea that, when adsorbed on isolating or semiconducting surfaces, nucleobases (precursors of terrestrial biomolecules) have an increased resistance against UV radiation (do not fragment), the underlying material playing an important role in dissipating energy and avoiding molecular fragmentation. Photostability studies of nucleic bases are especially relevant to an understanding of the lifetime and abundance of these molecules in space. These studies could, therefore, shed light onto prebiotic chemistry reactions. Mineral surfaces may play a relevant role to concentrate molecules, even under extremely diverse environments (UHV, solution, UV), which does not inhibit molecule/mineral surface interaction. Mimicking the complex prebiotic geochemical environment is still an enormous challenge; nevertheless, different environmental conditions could help to discriminate under which conditions molecule/mineral interactions would or would not be negligible.

We have used a PASC simulation chamber to investigate how the current atmospheric conditions of Mars with high UV radiation levels could affect the stability of different samples [77], some of them, samples containing olivine and pyrite. Our experimental results showed that the photocatalytic efficiency of pyrite leads to the formation of iron oxides-bearing minerals and sulphates under current Martian surface conditions. These results help to explain the unsuccessful detection of disulfide deposits on the Mars surface to date, which could be hidden by these secondary products [78,79]. We are currently investigating whether the exposure to different environmental conditions can compromise the capacity of specific clays to protect organic compounds under present-day Martian UV flux conditions, inside the Planetary Atmosphere and Surface Chamber (PASC) at CAB. In these experiments the Raman spectrometer implemented in the chamber is used for in situ monitoring the organic signal present in the mineral matrix [80].

Simulation chambers are essential tools for the assessment of the survival/adaptation of biological organisms under the harsh planetary conditions [81,82]. Studies of the survival capacity of lichens (eukaryotic and prokaryotic symbiotic organisms) under the conditions Mars surface conditions and space conditions have been performed in PASC. We have investigated the resistance of a symbiotic organism under simulated Mars conditions, exemplified with the lichen *Circinaria gyrosa*—an extremophilic eukaryote. We report the high resistance and survival capacity of a eukaryotic symbiotic organism under simulated Mars and space conditions. Our results show that neither any of the Martian atmosphere and surface UV climate combinations nor low-Earth orbit (LEO) vacuum conditions induce a significant decrease in lichen’s activity after an exposure of 120 h [82]. It is important to highlight that the thalli present in level-1 were exposed without any protection to simulated Mars and space conditions, especially unfiltered solar UV radiation, the most deleterious factor of space, which is directly absorbed by the DNA. PASC is a valid method to test the resistance potential of extremophile organisms under diverse harsh conditions and, thus, assess the habitability of extraterrestrial environments.

Additionally, an acidophilic chemolithotroph isolated from Río Tinto and *Deinococcus radiodurans* microorganisms were exposed to simulated Mars environmental conditions in PASC under the protection of a layer of ferric oxides and hydroxides, a Mars regolith analogue. Samples of these microorganisms were exposed to UV radiation in Mars atmospheric conditions at different time intervals under the protection of 2 and 5 mm layers of oxidized iron minerals. Viability was evaluated by inoculation on fresh media and characterization of their growth cultures. It was concluded that the presence of a thin layer of Mars regolith is critical to significantly reduce radiation doses and provide a shielding layer for microorganisms. Habitability increases considerably under only a few millimetres of regolith protection [81]. In conclusion, this experiment expanded the range of possibilities for extant life on Mars, meaning that a thin Mars regolith layer is enough to significantly reduce radiation doses and offers a shielding layer for these micro-organism species.

We have also presented evidence that the germinability of spores of the moss *Funaria hygrometrica* persists after exposure to simulated Martian environmental conditions inside PASC. The experimental results obtained using two species of moss, which demonstrate that both the spores of the moss *Funaria hygrometrica*, as well as the desiccated vegetative gametophyte shoots of the moss *Tortella squarrosa* (*Pleurochaete squarrosa*) were capable of resisting Simulated Martian Environmental Conditions (SMEC): Mars simulated atmospheric composition 99.9% CO_2_, and 0.6% H_2_O with a pressure of 7 mbar, 200 K and UV irradiation of 30 mW cm^−2^ in a wavelength range of 200–400 nm under a limited short time of exposition of two hours [83].

Therefore, the PASC planetary simulation chamber is an accurate experimental platform for contributing to the next planetary and space missions, to design successful scientific experiments under planetary environments to be further performed under real (in situ mission) planetary bodies, and to validate in situ measurements from orbital or rover observations on planets and moons surface. This unique simulation chamber has been part of the Europlanet facilities, has contributed and will reach new achievements of complementary knowledge regarding mineralogy, habitability conditions and survival/adaptation aspects of microorganism at extreme planetary conditions (Figure 5).

### 3.3. Deep Environments

The availability of cryogenic high pressure chambers opens the possibility to study deep environments, in particular aqueous reservoirs that are concealed under or within the icy crusts in the outer solar system moons (Figure 6). Raman spectroscopy is the main technique implemented because its function to detect particular ices such as gas clathrates and organics, differentiate between different phases of the same compound, and the preference to include this technique in future landing space missions [84,85] and into the laboratory simulation facilities. The high pressure consortia in Spain, named MALTA-team (Matter at High Pressure), has pushed interesting initiatives to develop new devices for planetary investigations and improvements in the chambers (http://www.malta-consolider.com [86]). 

There is no direct data of the aqueous internal layers so far, however, geophysical modelling of the icy moons from space mission’s data establish their possible global distribution and ranges of pressure and temperature for experimental research. Aqueous fluids are pressurized up to dozens MPa when in contact with silicates in the case of Europa and Enceladus, and to more than 1 GPa in the larger moons Ganymede and Titan, where the oceans are sandwiched between icy layers of different water phases. Several chemical systems have been suggested from indirect evidences, which include antifreeze components like salts, ammonia or methanol. As commonly happen in terrestrial studies, experimental (cryo) petrology help to understand the general properties and dynamics of these interior layers.

In active icy bodies, such as Europa and Enceladus, endogenous materials may reach the surface by different geological mechanisms. During ascent, aqueous fluids under pressure and at low temperatures might take with them clues from the interior environment, while modifying their properties. Examples of these investigations simulating the planetary interior are the characterization of the fractional precipitation and its consequences on the pH of endogenous brines [87], the quantification of the gas solubility in salty aqueous fluids for modelling cryovolcanic processes [88], the determination of the petrological evolution of interesting mineral assemblages and the geological effects on the surface [89], the evaluation of the salting-out caused by gas clathrate crystallization in fluids with salts and gases dissolved [90], or the determination of the guest-effect on high pressure phases of gas clathrates of Ganymede’s and Titan’s oceans [91,92].

The slow cooling of aqueous solutions composed by salts, like sulphates and carbonates, and also, sulphuric acid, led to the formation of mineral phases with a wide variety of hydration states, each one with different characteristics and behaviour at the icy crust. Fractional precipitations were done up to 30 MPa, no noting mineral phases changes respect to the experiments at 0.1 MPa. Along the differentiation, we calculated the variation of the pH of the liquid part performing the speciation of the system measured by Raman spectroscopy. In all the studied cases, the system suffers an acidification after the salt precipitation [87]. This simulation allows to make predictions about internal composition of icy crust of Europa, from the knowledge of the minerals found at the surface, directly measured by space probes like Galileo [93,94]. 

We used Raman spectroscopy to quantify the CO_2_ that is able to solubilize in salty aqueous solutions before crystallize at pressures up to 6 MPa (down to 5 km deep within Europa, assuming a crust mainly composed by water ice). The increase in temperature and salt concentration hinders the gas solubilisation. In our experiments we have demonstrated that the presence of gases resolves the buoyancy problem of the cryomagmas respect to an icy crust mainly composed by water ice. Furthermore, if the content of volatiles is high enough, our calculations show that even explosive phenomena would be possible [88], as the observed in Enceladus, Triton and, potentially, in Europa [95,96,97].

Water ice, gas clathrates and hydrated salts are potential candidates to play an important role in the geochemistry of the crust of icy satellites. We have studied the effects of having different proportions of the minerals in the endogenous assemblages [89]. We have recorded the volume changes that occur when the systems evolve by heating. Melting of internal chambers can happen from several sources, like thermal evolution, tidal forces, tectonic processes and/or radiogenic decay. In our experiments we show that the volume changes occurred during the melting of the different systems studied, are related to the ratio between the minerals of different type, and can explain the observation of several surface features with positive or negative relief respect to the surrounding (i.e., pits or domes, respectively).

The crystallization of gas clathrates in salty aqueous solutions, as the salt does not form part of the crystalline structure, promotes the concentration of the salt in the remaining aqueous solution. This process can modify severely the system, since if it is achieved the saturation level of the salts, it will occur their precipitation at higher temperatures than previously expected. Furthermore, different type of salt (with a different hydration level) may precipitate if the initial concentration is changed. The expulsion of the salt molecules after water-rich mineral (water ice and gas clathrates) formation is known as “salting-out” effect and it implies differentiation processes in aqueous environments. The process was simulated in our facilities. The MPPEC, with a large sapphire window, has allowed to quantify the process, that determine the layering than takes place and to perform a textural analysis of the diverse structures formed. At temperatures down to 267 K, CO_2_-clathrates float over the salty aqueous solution that derives in a later precipitation of hydrated salts [90,91,92,93,94,95,96,97,98]. Figure 7 is a good example of a run performed with respect to this issue.

CH_4_- and CO_2_-clathrates have been studied deeply by the combination of experimental studies with theoretical calculations. The type and the level of saturation of guests at the clathrate structure are an important parameter that influence in its stability properties, additionally to the pressure-temperature values. Experimental Raman spectra up to 1 GPa have been well-addressed by comparison with theoretical Raman spectra where guest-host interactions and cage deformations have been taken into account [91,92].

## 4. Conclusions

Future space and planetary missions will be motivated by ground-breaking technology and challenging scientific experiments, the achievement of which will be ensured through preliminary investigations in laboratory facilities. In this context, simulation chambers will play a fundamental role in garnering complementary knowledge and exploring potential difficulties before such space missions are designed and launched. Therefore, simulation chambers are an ideal and accurate tool for a large number of planetary exploration of habitable environments studies, as it has been reviewed and described above and, furthermore, will contribute to future space research. In addition, several studies previously described confirm the significance of ISAC, PASC and HPPEC simulation chambers as established instruments to emulate interstellar medium, planetary surfaces and deep environments. Furthermore, different multidisciplinary astrobiological studies can be assessed, which could contribute to evaluate the potential habitability of different environments towards the origin of life. 

## Figures and Tables

**Figure 1 life-09-00072-f001:**
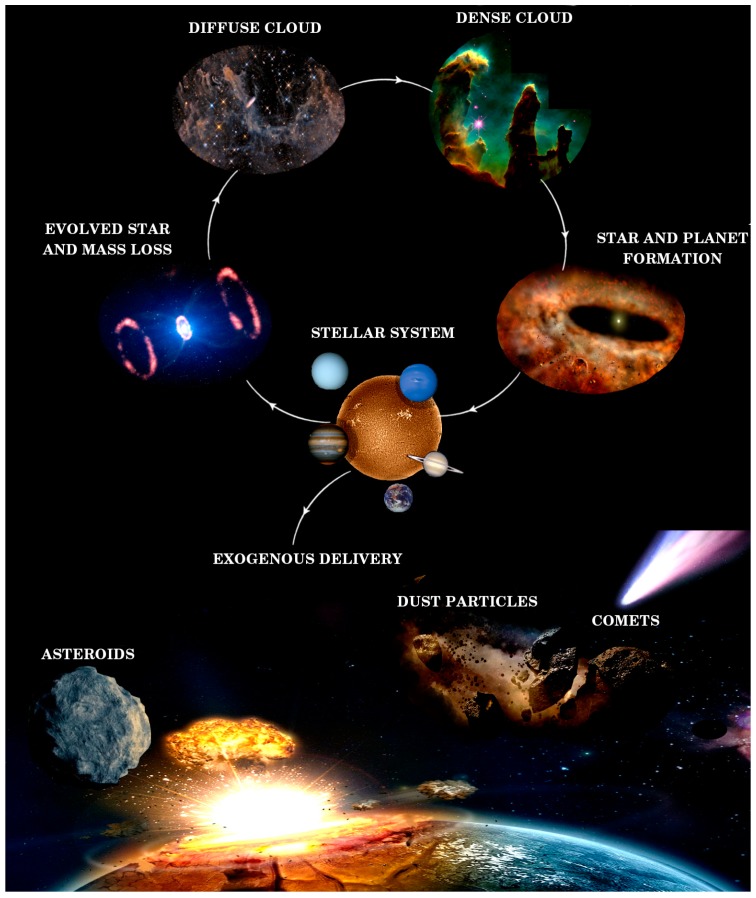
The cycle of matter in space and the delivery of exogenous material to the primitive Earth. Figure courtesy of G. A. Cruz Díaz using NASA images.

**Figure 2 life-09-00072-f002:**
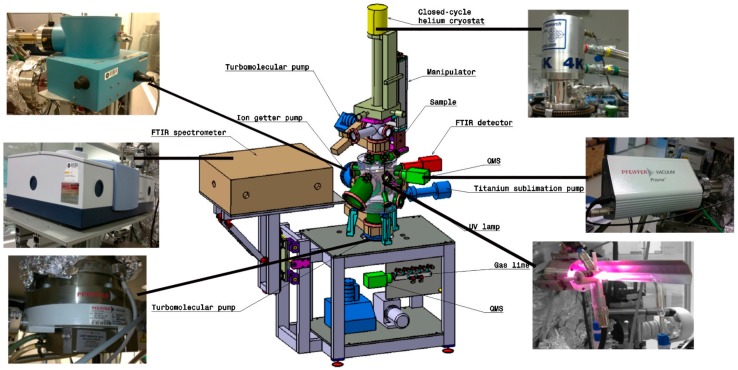
Schematic view of ISAC setup for the simulation of inter- and circumstellar ice processes. This image was adapted from [16] and picture additions by G. A. Cruz Díaz.

**Figure 3 life-09-00072-f003:**
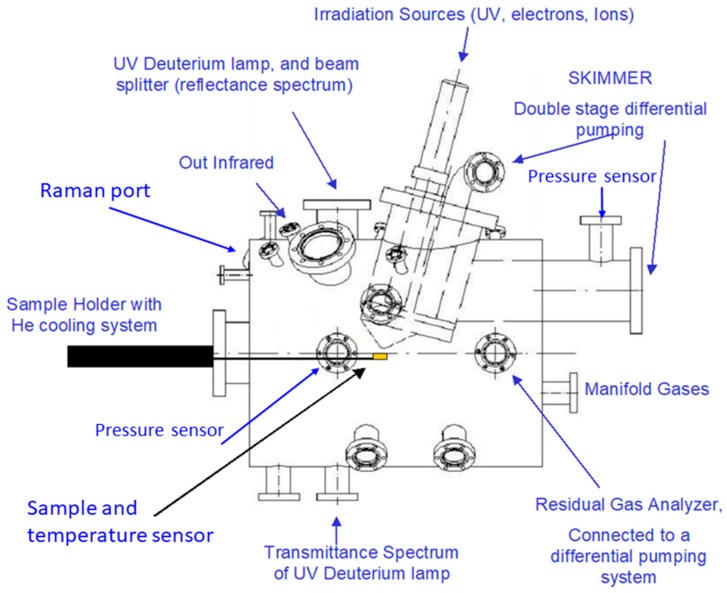
Schematic view of PASC design and setup for the planetary simulation environments.

**Figure 4 life-09-00072-f004:**
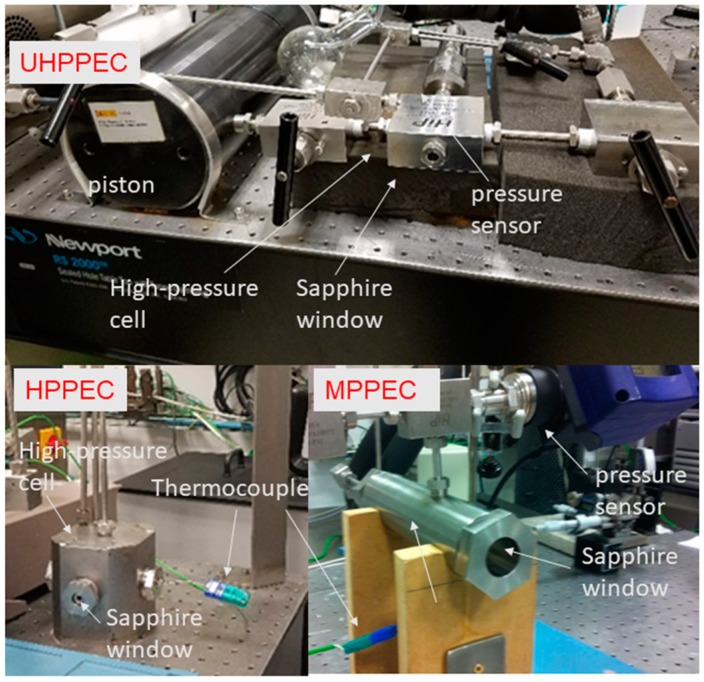
High-pressure chambers equipped with synthetic sapphire windows and pressure-temperature sensor for the physical-chemical characterization of the texture of the samples and by Raman spectroscopy.

**Figure 5 life-09-00072-f005:**
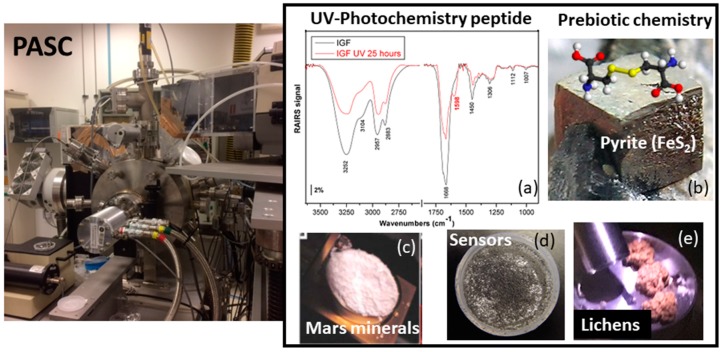
Photograph of the Planetary Atmosphere and Surfaces Chamber (PASC), and astrobiological applications of PASC chamber [6]: (**a**) Infrared spectra of peptide molecule (IGF) under the UV-photochemistry process; (**b**) prebiotic chemistry showing cystine biomolecule adsorption on a pyrite surface; (**c**) Mars minerals show perchlorate deliquescence process that may occur on Mars; (**d**) Near-UV irradiation transmittance studies as a function of the mass and thickness of basalt dust on Mars has been simulated; and (**e**) habitability studies show lichens due to their survival capacity.

**Figure 6 life-09-00072-f006:**
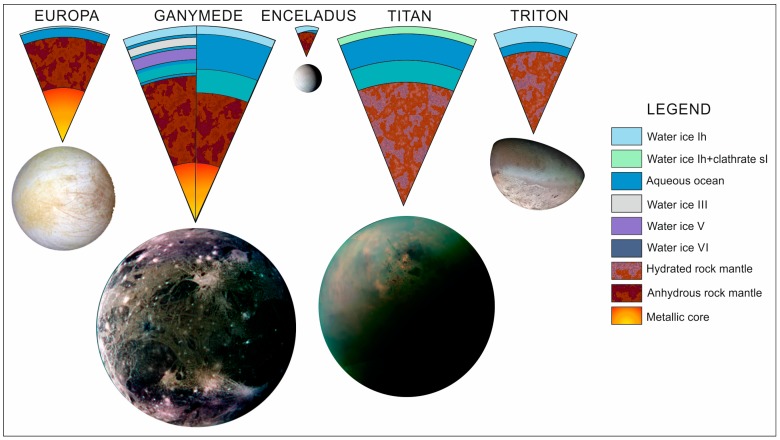
Internal structure of several ocean worlds of the solar system (Europa, Ganymede, Enceladus, Titan and Triton). Modified from Prieto-Ballesteros et al. 2014 [85].

**Figure 7 life-09-00072-f007:**
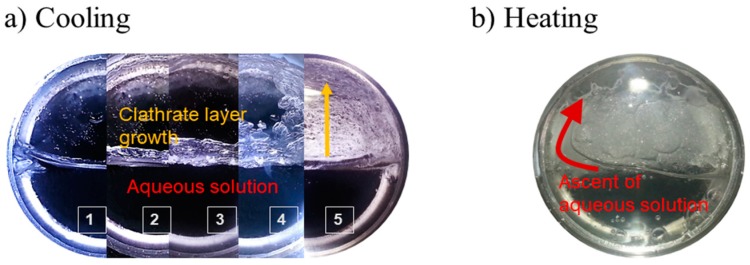
(**a**) Laminar clathrate growth from a salty aqueous solution saturated in CO_2_, at 3 MPa, when the system is cooled (**1**: 278 K, t = 0; **2**: 275 K, t = 12 h; **3**: 273 K, t = 24 h; **4**: 271 K, t = 36 h; **5**: 269 K, t = 48 h). (**b**) A later heating promotes the ascent of the gassy fluid through the upper crystal layer previously stabilized (the red curved arrow would indicate the ascent of the aqueous solution).

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
