# Peer review of "Characterizing Interstellar Medium, Planetary Surface and Deep Environments by Spectroscopic Techniques Using Unique Simulation Chambers at Centro de Astrobiologia (CAB)"

_life, 2019, doi:10.3390/life9030072_

Round 1

Reviewer 1 Report

The paper titled:"Characterizing interstellar medium, planetary surface and deep environments by spectroscopic techniques using unique simulation chambers at CAB", describes and reviews the results and the use of CAB (Centro de Astrobiologia) environmental simulation chambers in reproducing the behaviour of alien and weird environments.
This paper, also if it is not a strictly speaking research paper, is a very interesting review on the results of this special type of laboratory equipments and on their potentiality for future research in astrobiology field, comparative planetology and interstellar medium studies. In particular, I found adequate the chambers description and very clear the results section.

The paper is well written and easy to read.

I recommend the paper for the publication after that the following issues are addressed:

Major

- sec.1 Introduction:
The introduction to the paper describes well the rationale and the framework in which this review is loacated. Too bad,there are no references to any important work or review that can help the interested but not expert reader to better understand the described topics. In fact, the authors describe problems and processes of UV and X -rays interaction with the interstellar medium without any references. They also mention Solar System exploration space missions with no references to any paper that describe them, the same for habitability and other astrobiological concepts.
I suggest to revise the introduction adding references to explicative works where it is necessary and indicating reviews of the different topics.

-Figures: all the pictures in the figures, but figure 2, are at very low definition and of hard reading. But there are also other considerations, in particular with Figure 3, Figure 4 and Figure 6:

- Figure 3
In this Figure the three High Pressure chamber are shown. The picture are too small and the details are not visible. Moreover, on the contrary of the previous two (Figure 1 and Figure 2) no indications on the main points are given. E.g. where is the Raman spectroscopy port? and what about the pressure - temperature sensor? Please, enlarge the pictures indicating the important elements of the chambers.

- Figure 4.
This figure should show the PACS Chambers and its astrobiological applications. The pictures that should testified the astrobiological applications are too small and indecipherable. Furthermore their description into the caption does not help. I suggest to try to enlarge the pictures and identify each of them with a letter reported eventually in the caption with the description.

- Figure 6.
Also in this case the Figure is not clearly explained. Are these pictures taken through the window of the chamber? It seems that the A panel of the figure would like to present an evolution in time of the clathrate layer growth. Could the authors identify clearly the time stamps of the growth and the temperature of each picture? Panel B is also not so clear. e.g. what the curved arrow would indicate?

Minor
- Please specify each acronyms in the text.

Author Response

We are pleased to know that the manuscript has been reviewed, although there are some issues to clarify in the paper. We have considered the referee’s comments helpful to produce an improved version of the final manuscript. We are especially pleased about the fact that the two referees have emphasise the relevance of these studies in the astrobiology field and the interest for the wider astrobiology community, which is the aim of our scientific contribution in this manuscript.

We include a new version of the manuscript incorporating these changes (blue highlighted) that we expect it will be suitable for publication. Here after follows our response to the comments raised by the referees and a list of the modifications introduced in the revised manuscript.

 Reviewer 1 Comments and Suggestions for Authors

I recommend the paper for the publication after that the following issues are addressed:

Major

- sec.1 Introduction:
The introduction to the paper describes well the rationale and the framework in which this review is located. Too bad, there are no references to any important work or review that can help the interested but not expert reader to better understand the described topics. In fact, the authors describe problems and processes of UV and X -rays interaction with the interstellar medium without any references. They also mention Solar System exploration space missions with no references to any paper that describe them, the same for habitability and other astrobiological concepts.
I suggest to revise the introduction adding references to explicative works where it is necessary and indicating reviews of the different topics.

We agree with the reviewer, therefore we have improved the introduction section including 10 new references (highlighted in blue in the references section), which are related to important work or review that can help the interested but not expert reader to better understand the described topics. Also, we have included references related to Solar System exploration space missions. Furthermore, we have added a new figure 1 which explains and illustrates the cycle of matter in space and the delivery of exogenous material to the primitive Earth, to clarify processes of UV and X -rays interaction with the interstellar medium as the reviewer suggested.

-Figures: all the pictures in the figures, but figure 2, are at very low definition and of hard reading. But there are also other considerations, in particular with Figure 3, Figure 4 and Figure 6 (New figures 4, 5 and 7):

- Figure 3
In this Figure the three High Pressure chamber are shown. The picture are too small and the details are not visible. Moreover, on the contrary of the previous two (Figure 1 and Figure 2) no indications on the main points are given. E.g. where is the Raman spectroscopy port? and what about the pressure - temperature sensor? Please, enlarge the pictures indicating the important elements of the chambers.

Taking in account the reviewer comments, we have modify and improve the old Figure 1, 2 and 3 (new figures 2, 3 and 4 respectively in the new version of the manuscript): New figure 4 has been re-done to make it bigger and visible the details on it. In new figure 3 Raman port, temperature and pressure sensor has been identify. Moreover, quality and definition for the three figures have been improved to make easily reading.

- Figure 4 (New Figure 5)
This figure should show the PACS Chambers and its astrobiological applications. The pictures that should testified the astrobiological applications are too small and indecipherable. Furthermore their description into the caption does not help. I suggest to try to enlarge the pictures and identify each of them with a letter reported eventually in the caption with the description.

Following the reviewer suggestion old figure 4 (new figure 5) has been highly modify in order to clarify and make easily readable the different astrobiological applications. Furthermore, the figure 5 caption has been re-written in order to complete the description of each process and help to identify each application with (a), (b)… labels in the figure.

Figure 5 Photograph of the Planetary Atmosphere and Surfaces Chamber (PASC), and Astrobiological applications of PASC chamber [6]: (a) Infrared spectra of peptide molecule (IGF) under UV-Photochemistry process; (b) Prebiotic chemistry show cystine biomolecule adsorption on a pyrite surface; (c) Mars minerals show perchlorate deliquescence process that may occur on Mars; (d) Near-UV irradiation transmittance studies as a function of the mass and thickness of basalt dust on Mars has been simulated; and (e) Habitability studies show lichens due to their survival capacity.

- Figure 6 (New Figure 7).
Also in this case the Figure is not clearly explained. Are these pictures taken through the window of the chamber? It seems that the A panel of the figure would like to present an evolution in time of the clathrate layer growth. Could the authors identify clearly the time stamps of the growth and the temperature of each picture? Panel B is also not so clear. e.g. what the curved arrow would indicate?

Following the reviewer suggestion old figure 6 has been modify to the new version Figure 7, and the missing information has been included in the figure 7 caption, in order to clarify and to explain the concerns raised by the reviewer as a follow:

Figure 7. a) Laminar clathrate growth from a salty aqueous solution saturated in CO2, at 3 MPa, when the system is cooled (1: 278 K, t = 0; 2: 275 K, t = 12 h; 3: 273 K, t = 24 h; 4: 271 K, t = 36 h; 5: 269 K, t = 48 h). b) A later heating promotes the ascent of the gassy fluid through the upper crystal layer previously stabilized (the red curved arrow would indicate the ascent of the aqueous solution).

Minor
- Please specify each acronyms in the text.

The manuscript has been revised and the acronyms have been specify. The corrections have been highlighted in blue in order to find them easily.

Thank you very much for your help,

Sincerely

Eva

Reviewer 2 Report

This manuscript is a review that describes all the facilities used by authors to simulate the interstellar medium, the planetary surfaces, and deep environments. For interstellar medium simulation, the ultra-high vacuum chamber ISAC has been used. This setup allowed several experimental investigations including the study of the UV photodesorption of CO ice and other molecular ices, the measurement of the absorption cross-section of several ice species, and the study of the sulfur chemistry in the ice. Planetary surfaces have been simulated using the ultra-high-vacuum PASC simulation facility coupled to Raman and infrared spectroscopy. PASC has been used for example to study the effect of perchlorates salts on the Mars surface, the stability of samples containing olivine and pyrite irradiated with UV photons similar to the Mars environment, and the interaction of amino acids, small peptides and nucleic bases on several mineral surfaces. Finally, deep environments have been simulated using the cryogenic medium, high, and ultra-high -pressure Planetary Environment Chamber coupled to Raman spectroscopy. Among the experimental results obtained with these setups, the fractional precipitation study and the measurement of the subsequent variation of pH of the liquid part and the quantification of the soluble CO2 fraction in salty aqueous solutions. Results obtained in the high-pressure chambers showed also that the ratio between different minerals can affect the volume changing during the melting process which can explain several surface features in Jupiter’s Europa satellite. The Medium-Pressure Planetary Environment Chamber has been used to simulate the “salting out” process in aqueous environments and study the subsequent structure evolution of clathrate-salt hydrate.

While this paper didn’t “report results that have not been submitted or published before, even in part” as required from manuscripts submitted to MDPI journal, it provides an exhaustive summary of the astrobiological related studies performed with the described facilities. Such paper is important, in my opinion, to highlight the important role of laboratory simulation to complement space-based missions and help the interpretation of its return observations. The editor will have the final decision regarding the acceptance or rejection of the manuscript.

Minor modification:

The introduction of this paper didn’t include any bibliographic references. Many scientific statements should be revised to include references (exp. line 40, 43, 44…) This paper is a review for the astrobiological related results obtained with the described facilities. I encourage authors to cite similar facilities that have the same research applications in the introduction. I encourage the authors to clearly present the goal of the manuscript as a ‘review’ of the experimental research conducted with the described facilities in the abstract, introduction and conclusion. The title of the section “results” can be improved as well and replaced by “review of experimental results” for example. Since this manuscript is a review paper, I encourage the authors to provide an exhaustive description of the experimental facilities including the in-situ diagnostics and the experimental challenges. The reader will not need to read another reference to have the full description of the setups. I encourage the authors to add if possible figures with the key results especially in the interstellar and planetary surface simulation sections. Minor spell checks need to be addressed:

Line 72-73 sentence is missing the verb

Line 135: UV-Vis irradiation

Line 269: mineral surfaces.

Line 350-358: needs to be restructured

Line 360 possible

Line 415: that

Line 426 is->are

Authors should consider using short sentences and remove many statements between comma.

Authors need to remove commas that separate the subject from the verb

Author Response

We are pleased to know that the manuscript has been reviewed, although there are some issues to clarify in the paper. We have considered the referee’s comments helpful to produce an improved version of the final manuscript. We are especially pleased about the fact that the two referees have emphasise the relevance of these studies in the astrobiology field and the interest for the wider astrobiology community, which is the aim of our scientific contribution in this manuscript.

We include a new version of the manuscript incorporating these changes (blue highlighted) that we expect it will be suitable for publication. Here after follows our response to the comments raised by the referees and a list of the modifications introduced in the revised manuscript.

Reviewer 2 Comments and Suggestions for Authors

Minor modification:

The introduction of this paper didn’t include any bibliographic references. Many scientific statements should be revised to include references (exp. line 40, 43, 44…). This paper is a review for the astrobiological related results obtained with the described facilities. I encourage authors to cite similar facilities that have the same research applications in the introduction. I encourage the authors to clearly present the goal of the manuscript as a ‘review’ of the experimental research conducted with the described facilities in the abstract, introduction and conclusion. The title of the section “results” can be improved as well and replaced by “review of experimental results” for example. Since this manuscript is a review paper, I encourage the authors to provide an exhaustive description of the experimental facilities including the in-situ diagnostics and the experimental challenges. The reader will not need to read another reference to have the full description of the setups. I encourage the authors to add if possible figures with the key results especially in the interstellar and planetary surface simulation sections. Minor spell checks need to be addressed:

We agree with the reviewer, therefore we have improved the introduction section including 10 new references (highlighted in blue in the references section) related to important work or review of similar facilities that have the same or similar research applications.

Following the reviewer comments, we have included in the abstract the following sentence: “A review of many wide-ranging applications in astrobiology are detailed herein to provide an understanding of the potential and flexibility of these unique experimental systems”. Furthermore, the title of the section “results” has been replaced by “review of experimental results” as the reviewer suggested. And finally, we have included in the conclusions section the following sentence: “Therefore, simulation chambers are an ideal and accurate tool for a large number of planetary exploration of habitable environments studies, as it has been reviewed and described above, furthermore will be contributing to future space research”, to clearly present the goal of the manuscript as a ‘review’ of the experimental research conducted with the described facilities in the abstract, experimental results and conclusion sections.

A new Figure 3,4 and 5 have been done in order to clarify set-ups of the experimental systems and experimental challenges and results.

Line 72-73 sentence is missing the verb

The sentence has been rephrased as: The European mission ExoMars with the TGO orbiter and EDM lander, is used as precursor mission for the 2020 ExoMars (Rosalind Franklin) rover and surface platform.

Line 135: UV-Vis irradiation: It has been corrected

Line 269: mineral surfaces. It has been corrected

Line 350-358: needs to be restructured.

 The paragraph has been restructured in order to clarify the meaning as following:

“The availability of cryogenic high pressure chambers open the possibility to study deep environments, in particular aqueous reservoirs that are concealed under or within the icy crusts in the outer solar system moons (Fig. 6). Raman spectroscopy is the main technique implemented because its function to detect particular ices such as gas clathrates and organics, differentiate between different phases of the same compound, and the preference to include this technique in future landing space missions [85-86] and into the laboratory simulation facilities. The High Pressure consortia in Spain, named MALTA-team (Matter at High Pressure), has pushes interesting initiatives to develop new devices for planetary investigations and improvements in the chambers (http://www.malta-consolider.com, [87]).“

Line 360 possible: It has been corrected

Line 415: that. It has been corrected

Line 426 is->are. It has been corrected

 Thank you very much for your help,

Sincerely

Eva

Round 2

Reviewer 1 Report

Authors have fulfilled all the requests (thank you) and I have no more issue to highligth about this interesting paper.

I suggest to publish it without any other reviews.

Best regards.